# Effects of Different Generations and Sex on Physiological, Biochemical, and Growth Parameters of Crossbred Beef Cattle by *Myostatin* Gene-Edited Luxi Bulls and Simmental Cows

**DOI:** 10.3390/ani13203216

**Published:** 2023-10-14

**Authors:** Chao Hai, Chunling Bai, Lei Yang, Zhuying Wei, Hong Wang, Haoran Ma, Haibing Ma, Yuefang Zhao, Guanghua Su, Guangpeng Li

**Affiliations:** 1State Key Laboratory of Reproductive Regulation and Breeding of Grassland Livestock, College of Life Science, Inner Mongolia University, Hohhot 010070, China; h15248037201@163.com (C.H.); chunling1980_0@163.com (C.B.); leiyang@imu.edu.cn (L.Y.); weizhuying2008@126.com (Z.W.); zhaoyf@imu.edu.cn (Y.Z.); 2Sheng-Quan Ecological Animal Husbandry Company, Chifeng 024500, China; wh793333@163.com; 3Institute of Animal Science, Chinese Academy of Agricultural Sciences, Beijing 100193, China; mahaorancaas@163.com; 4Inner Mongolia Aokesi Animal Husbandry Co., Ltd., Hesge Ula Ranch, Ulagai Management Area, Xilingol League 026321, China; mhb1971@126.com

**Keywords:** *Myostatin*, gene-edition, beef cattle, multiple linear regression, hematological parameters, biochemical parameters, growth parameters

## Abstract

**Simple Summary:**

The study described the effects of *Myostatin* gene editing on cattle to improve the growth and muscle development of hybrid cattle offspring. We focused on the *Myostatin* gene, which plays a crucial role in regulating muscle development. Gene editing could be able to enhance the size and performance of the crossbred cattle. The results showed that the *Myostatin* gene-edited cattle offspring displayed improved body size parameters, particularly body weight, cross height, and hip height, indicating the significant development of the hindquarters. The edited cattle have “double-muscled” forequarters and hindquarters with clear muscle masses, and their coat color reverted back to the original yellow. The study also found differences in blood glucose, calcium, and low-density lipoprotein levels, as well as a decrease in serum insulin levels in the edited cattle. Overall, the findings suggest that breeding with these *Myostatin* gene-edited bulls improve growth and performance in crossbred cattle, benefiting both farmers and consumers.

**Abstract:**

(1) Background: Myostatin (MSTN) is a protein that regulates skeletal muscle development and plays a crucial role in maintaining animal body composition and muscle structure. The loss-of-function mutation of *MSTN* gene can induce the muscle hypertrophic phenotype. (2) Methods: Growth indexes and blood parameters of the cattle of different months were analyzed via multiple linear regression. (3) Results: Compared with the control group, the body shape parameters of F2 cattle were improved, especially the body weight, cross height, and hip height, representing significant development of hindquarters, and the coat color of the F2 generation returned to the yellow of Luxi cattle. As adults, *MSTN* gene-edited bulls have a tall, wide acromion and a deep, wide chest. Both the forequarters and hindquarters are double-muscled with clear muscle masses. The multiple linear regression demonstrates that *MSTN* gene-edited hybrid beef cattle gained weight due to the higher height of the hindquarters. Significant differences in blood glucose, calcium, and low-density lipoprotein. Serum insulin levels decreased significantly at 24 months of age. *MSTN* gene editing improves the adaptability of cattle. (4) Conclusions: Our findings suggest that breeding with *MSTN* gene-edited Luxi bulls can improve the growth and performance of hybrid cattle, with potential benefits for both farmers and consumers.

## 1. Introduction

Myostatin is a vital negative regulator in the process of skeletal muscle development, which plays a critical role in maintaining animal body traits and overall muscle structure [1]. The *MSTN* gene preserves a highly conserved genetic structure among a wide range of species, including cattle, sheep, rabbits, dogs, humans, fish, pigs, chickens, baboons, and zebrafish. The homogeneity of its C-terminal sequence is estimated to be >90% [2]. A mutation within the *MSTN* gene directly impacts the structure of the secreted protein resulting in a double-muscular phenotype. Currently, there have been six functional mutation loci were identified in Belgian Blue Cattle [3], Piedmontese [3], Charolais [4], Limousin [5], Blonde d’Aquitaine [6], and German Gelbvieh [7]. Cattle that carry such a mutation will present with a muscular hypertrophic phenotype.

Simmental cattle, originating from Central Europe in the 1830s, have been globally recognized for their outstanding attributes in draught, meat, and milk production [8]. In order to enhance the indigenous cattle population in China during the 1950s, Simmental cattle were introduced. After years of selective breeding initiatives, Chinese Simmental cattle have exhibited significant genetic progress in both their phenotypes and genetics. Currently, Chinese Simmental cattle have become one of the major beef cattle breeds in China, known for their rapid growth and delicious meat quality, accounting for approximately 70% of the beef market [9]. Luxi cattle, an indigenous breed, have an excellent flavor and fragrance and good fat deposition, but with a relatively slow growth rate, underdeveloped hindquarters, and low meat production in carcasses. To address these shortcomings, we leveraged the natural mutation of *MSTN* and utilized CRISPR/Cas9 technology to conduct site-directed editing of the Luxi cattle *MSTN* gene. This resulted in the progeny of bulls with significantly improved body appearance and growth traits [10].

In this study, the *MSTN* gene-edited Luxi bulls were used to mate with Simmental cows to produce hybrids with Luxi cattle pedigree and *MSTN* mutation. There are two main purposes for this: firstly, to breed healthy homozygous *MSTN* gene-edited cattle offspring, and secondly, to improve Chinese native cattle breeds and enhance body weight and lean meat rate. On the other hand, linear regression is an important tool for statistical analysis. The growth performance and hematological and biochemical parameters of *MSTN* gene-edited hybrid beef cattle were analyzed via multiple linear regression. In brief, by using gene editing technology and gene editing products, combined with conventional crossbreeding methods, the double-muscle type cattle have improved meat quality from Chinese yellow cattle but also have the advantages of fast growth, good maternal ability, and the strong adaptability of Simmental cattle.

## 2. Materials and Methods

### 2.1. Animals

All Chinese Simmental cattle and the hybrid cattle were bred on the farm of Chifeng Shengquan Ecological Herding Co., which is located in Chifeng City, Inner Mongolia Autonomous Region, China (42°26′–43°25′ N, 117°43′–120°43′ E). The cattle were maintained at the ranch under standard feed and housing conditions. Feed composition and nutrient composition are shown in Appendix A. Animal were approved by the Institutional Animal Care and Use Committee at Inner Mongolia University (approval ID: IMU-CATTLE-2017-059).

The hybrid cattle were produced by mating Chinese Simmental cows with the *MSTN*^−/−^ Luxi bulls generated using the CRISPR/Cas9 gene editing system [11,12] through artificial insemination. We generated hybrid offspring by crossing two different genotypes of *MSTN*^−/−^ Luxi bulls (Appendix A). Genomic DNA was extracted from the calves, and PCR was employed to identify the DNA sequences of the *MSTN* gene. Sequence analysis revealed a deletion of 6 bp (g.507del (6)) at position 507 (Appendix A) or a deletion of 115 bp (g.505del (115)) at position 505 (Appendix A), corresponding to our statistics on the F1 generation of crossbred cattle (Appendix A). When F1 generation *MSTN*^+/−^ cows grew to 24 months of age, they were mated with *MSTN*^−/−^ Luxi bulls to produce F2 beef cattle. Crossbred progenies of F1 and F2 were evaluated from birth to 24 months (Figure 1). Dynamic weight and body size were measured from birth to 24 months. Blood physiological and biochemical parameters from 9 to 24 months of age were analyzed. For the same generation of cattle, we selected hybrid cattle within ±1 month of age for tracking measurements and blood tests.

### 2.2. Body Weight and Body Size Measurement

The study included 20 Luxi bulls, 25 Luxi cows (CON), 30 F1 generation bulls, 45 F1 generation cows, 45 F2 generation bulls, and 36 F2 generation cows. After the calves were born, birth weights and body sizes were measured. Values of body weight (BW), body oblique length, shoulder width, cross width, hip width, body height, cross height, hip height, chest circumference, abdominal circumference, and tube circumference were measured, respectively when calves and youth cattle grew to 3, 6, 9, 12, 15, 18, and 24 months of age (Figure 2).

### 2.3. Detection of Blood Physiological and Biochemical Parameters

Blood samples were collected at 9, 12, 15, 18, and 24 months of age, respectively, and 10 mL of blood was collected from the jugular veins into tubes containing ethylenediaminetetraacetic acid (EDTA) as an anticoagulant for hematological analysis, while blood samples were collected in serum separator tubes (SST) for biochemical analysis. Hematological examinations were performed within 24 h of blood collection. The blood samples were then analyzed by MSCAN-II dry analyzer to determine white blood cells (WBC), red blood cells (RBC), hemoglobin (HGB), lymphocyte percentage (LYM%), hematocrit (HCT), platelets (PLT), and basophilic granulocyte (BAS). The blood for serum was slowly poured into a 15 mL centrifuge tube and centrifuged at 3000 r for 10 min. The prepared serum was transferred into a 1.5 mL cryo-storage tube, labeled, and stored in liquid nitrogen. The biochemical parameters of alkaline phosphatase, glutamyl transferase, aspartate aminotransferase, alanine aminotransferase, blood amylase, urea nitrogen, glucose (Glu), alanine aminotransferase (ALT), aspartate aminotransferase (AST), total protein (TP), albumin (ALB), creatine kinase (CK), high-density lipoprotein cholesterol (HDL-C), low-density lipoprotein cholesterol (LDL-C), amylase (AMY), calcium (Ca), alkaline phosphatase (ALP), lactate dehydrogenase (LDH-L), bicarbonate (CO2-L), cholinesterase (CHE), cholesterol (CHOL), lactic acid (LACT), triglyceride (TRIGL), UREAL, lipase (LIPC), and creatinine (CRE) were determined using a Cobas c311 analyzer (ROCHE).

### 2.4. Determination of Serum Insulin

The serum insulin concentration was determined by enzyme-linked immunosorbent assay (Insulin, Bovine ELISA, Mercodia, Uppsala, Sweden). Samples of 25 μL sera from five 12-month-old cattle and five 24-month-old cattle were enzyme-conjugated; were incubated on a plate shaker (800 rpm) for 2 h at room temperature (18–25 °C); were washed 6 times with 700 µL wash buffer 1 × solution per tube; 200 μL Substrate TMB was added into each tube; were incubated on the bench for 15 min at room temperature (18–25 °C); 50 μL Stop Solution was added to each well and collected in 96-well plates; and optical density was read at 450 nm and results were calculated.

### 2.5. Correlation Analysis

All correlation analyses were performed using Spearman’s correlation. Correlation analysis was conducted using R software version 3.4.

### 2.6. Statistical Analysis

SPSS 25.0 software was used for analyzing the significance difference (one-way ANOVA was used for multiple group comparisons and Student’s *t*-test was used for two-group comparisons) and *p* < 0.05 was a significant difference, and *p* < 0.01 was extremely significant differences between the different groups.

The multiple linear regression was analyzed using R software version 3.4 or in the command line. Sex, age, and generations of the cattle were fitted as independent variables while growth, hematological, and biochemical parameters were fitted as response variables.

The multiple linear regression used for the analysis of biochemical parameters was
(1)$$y=b0+b1i+b2j+b3k+\left(b1\times b2\right)ij+\left(b2\times b3\right)jk+\left(b1\times b3\right)ik+\left(b1\times b2\times b3\right)ijk+\varepsilon$$
where *y* = the response variables (Glu, AST, ALT, TP, ALB, CK, HDL-C, LDL-C, AMY, Ca, ALP, LDH, CO2-L, CHE, CHOL, LACT, TRIGL, UREAL, LIPC, CRE); *b*0 = the intercept in multivariate linear regression; *b*1*i* = effect of sex (female, male); *b*2*j* = effect of age (9, 12, 15, 18, 24 months old); *b*3*k* = effect of generations (wild-type Luxi cows, F1 crossbred beef cattle, F2 crossbred beef cattle); (b1×b2)ij = interaction between sex and age; (b2×b3)jk = interaction between age and generations; (b1×b3)ik = interaction between sex and generations; and (b1×b2×b3)ijk = interaction between sex, age and generations. ε = represents the random error.

The multiple linear regression used for the analysis of hematological parameters was
(2)$$y=b0+b1i+b2j+b3k+\left(b1\times b2\right)ij+\left(b2\times b3\right)jk+\left(b1\times b3\right)ik+\left(b1\times b2\times b3\right)ijk+\varepsilon$$
where *y* = the response variables (WBC, LYM, RBC, HCT, PLT, BAS, HGB); *b*0 = the intercept in multivariate linear regression; *b*1*i* = effect of sex (female, male); *b*2*j* = effect of age (9, 12, 15, 18, 24 months old); *b*3*k* = effect of generations (wild-type Luxi cows, F1 crossbred beef cattle, F2 crossbred beef cattle); (b1×b2)ij = interaction between sex and age; (b2×b3)jk = interaction between age and generations; (b1×b3)ik = interaction between sex and generations; and (b1×b2×b3)ijk = interaction between sex, age, and generations. ε = represents the random error.

The multiple linear regression used for the analysis of growth parameters was
(3)$$y=b0+b1i+b2j+b3k+\left(b1\times b2\right)ij+\left(b2\times b3\right)jk+\left(b1\times b3\right)ik+\left(b1\times b2\times b3\right)ijk+\varepsilon$$
where *y* = the response variables (BW, shoulder height, cross height, hip height, shoulder width, cross width, hip width, chest circumference, abdominal circumference, body oblique length, tube circumference); *b*0 = the intercept in multivariate linear regression; *b*1*i* = effect of sex (female, male); *b*2*j* = effect of age (0, 3, 6, 9, 12, 15, 18, 24 months old; *b*3*k* = effect of generations (wild-type Luxi cows, F1 crossbred beef cattle, F2 crossbred beef cattle); (b1×b2)ij = interaction between sex and age; (b2×b3)jk = interaction between age and generations; (b1×b3)ik = interaction between sex and generations; and (b1×b2×b3)ijk = interaction between sex, age and generations. ε = represents the random error.

## 3. Results

### 3.1. Physical Appearance and Growth Parameters

Compared with the control Luxi cattle, F1 cattle showed an improvement in body size parameters. F2 generation cattle were further improved in weight and hindquarters, especially body weight, cross height/width, and hip height, representing the significant development of hind limb muscles. Their coat color returned to that of Luxi cattle yellow, and a few of the cattle had white faces (Figure 3).

The F1 and F2 cattle were superior to Luxi cattle in body weight. No significant difference between generations was due to the similar body weight of F1 and F2 cattle (*p* = 0.461). However, the interaction between age and sex is significant (*p* < 0.001). Cross height saw significant differences between generations (*p* = 0.028), and there moderate differences were observed in hip height, although this was non-significant (*p* = 0.057) between generations (Table 1). The results showed that the hybrid cattle were overall improved in body shape, and significant differences were observed in hindquarter height among different generations. The development degree of the hindquarters of hybrid cattle improved significantly from generation to generation. Tube circumference values have significant interactions between sex, age, and generation.

### 3.2. Hematological Parameters

Hematological parameter data and interactions between sex, age, and generation are shown in Table 2. The hematological parameters were significantly affected by age except for HCT, BAS, and HGB. WBC and RBC were significantly affected by sex. WBC, LYM (%), and RBC exhibited a significant age-related decrease (*p* < 0.05). Furthermore, there were significant differences in LYM (%) and RBC among different generations (*p* < 0.05). In addition, LYM (%) demonstrated significant interactions between age and generation, and RBC has significant interactions between gender and generation; age and generation; and gender, age, and generation. The results show that although some of the routine blood indicators were different between generations, they were within the normal ranges. LYM (%) decreased significantly in generations and ages, which might be related to weight gain. LYM (%) of the F1 hybrid cattle was slightly higher than that of F2.

### 3.3. Biochemical Parameters

Most biochemical indicators do not differ significantly between generations, and all indicators are within the normal ranges. Our results showed that Glu, ALB, LDL-C, CO2-L, CHE, and UREAL were significantly affected by generations (*p* < 0.05, Figure 4). The blood Glu of F1 and F2 cattle were significantly lower than WT, but no significant interaction with age and gender. There were significant differences in LDL-C and Ca in age, gender, generation, and the interactions between them and low level of LDL-C in hybrid cattle. Conversely, higher levels of HDL-C were found in hybrid cattle, although there was no significant differences between generations (*p* = 0.073). Total cholesterol and triglyceride levels were lower in hybrid cattle. LACT in hybrid cattle was significantly higher than that in the control group (Table 3).

### 3.4. Relationship between BW and Hematological, and Biochemical Parameters

Given the consistent body weight advantage observed in both F1 and F2 hybrid cattle across nearly every age group, we were interested in investigating the potential correlation between blood parameters and body weight. Then, body weight and hematological parameters and biochemical parameters were compared using Spearman’s correlation. By comparison, the correlation between both BW and hematological parameters was low and most show a negative correlation (R < 0.4), LYM% showed the highest negative correlation (R = −0.39, Figure 5A, Appendix A). Contrary to blood hematological parameters, more blood biochemical parameters show a positive correlation with BW. Ca and BW showed the strongest correlations. In addition to Ca, AMY, and LDL_C also showed a high positive correlation. Glu had the highest negative correlation with body weight (Figure 5B, Appendix A). The results indicated that most of the blood parameters exhibited a low correlation with body weight. The high correlation between Ca, Glu, and body weight could reflect the differences between hybrid cattle and the control group.

### 3.5. Determination of Serum Insulin

The notable variations in biochemical indicators, such as serum Ca, particularly in relation to insulin, among different generations prompted our investigation. We collected five blood serum samples from fasting cattle to assess insulin levels using ELISA. The findings revealed that, at 12 months of age, there was no significant disparity in serum insulin levels between Luxi cattle and *MSTN* gene-edited hybrid cattle. However, by the age of 24 months, there was a noticeable overall increase in serum insulin levels. Notably, the serum insulin levels of the F1 and F2 generations of *MSTN* gene-edited hybrid cattle were significantly lower than those of Luxi cattle, as depicted in Figure 6. These results suggest that *MSTN* may sustain lower insulin levels for an extended period.

## 4. Discussion

*Myostatin* gene plays a crucial role in animal growth, development, and body pattern formation. Mutations in the *MSTN* gene not only impacts skeletal muscle growth and development but also influences physiological processes such as glucose metabolism, lipid metabolism, and protein metabolism [1]. *MSTN* mutations have been identified in various species, including dogs [13], sheep [14], goats [15], and humans [16], all of which exhibit enhanced muscular development. Through the use of gene-editing technologies, researchers have successfully generated *MSTN* gene-edited animals, including sheep [17], cattle [18], pigs [19], dogs [20], rabbits [21], and goats [22,23] have been successively obtained. However, there is limited information available regarding the utilization of *MSTN* gene-edited animals for breeding improvement and crossbreeding purposes.

Mutation in the *MSTN* gene leads to a hypertrophic muscle phenotype in cattle, resulting in significant improvements in the slaughter rate and meat yield of beef cattle. Crossbreeding ordinary cows with *MSTN*-mutated Belgian Blue and Piedmont bulls led to a substantial increase of 20% to 25% in muscle mass in the hybrid cattle [2]. The live weight, carcass weight, and slaughter rate of the crossbred Belgian Blue bulls and Holstein cows were significantly higher compared to normal breeding cattle. Domingo et al. [24] found that the slaughter rate of 10-month-old Belgian blue × Holstein cows was 59.44 ± 1.22%, significantly higher than that of Limousin × Holstein cows (55.17 ± 1.26%) and Rubia Gallega × Holstein cows (54.86 ± 1.43%). Similarly, Keady et al. [25] crossbred Belgian blue and Angus bulls with Holstein cows and slaughtered them at 299 d, respectively. The average carcass weight of Belgian blue × Holstein crossbred cattle was 369 kg, which was significantly higher than that of Angus × Holstein crossbred cattle (354 kg). Compared with normal cattle, *MSTN* hybrid cattle had a larger eye muscle area, lower back fat, higher slaughter rate, and lower fat content ratio [26]. In this study, Chinese Simental cows were crossed with *MSTN* gene-edited Luxi bulls, the hybrid generations carried a haplotype mutated *MSTN*. Compared with the control Luxi and Simmental cattle, all the growth traits of *MSTN*-gene offspring were improved, especially body weight, chest circumference, and hip height/weight representing hindquarters development. After adulthood, the hybrid bulls had high and broad acromions and deep and wide chests. Both forequarters and hindquarters were double-muscled with clear muscle masses. The back was straight, the middle body was muscular, and the body was rectangular. The results of multiple linear regression analyses are presented in this study. The results indicate significant difference in body weight between generation and age but no significant differences between generations. The possible reason was that although there is no significant difference in body weight between F1 and F2 generations, the growth rate of body weight varies with age. *MSTN* mutation could improve the performance of F1 and F2 crossbred beef cattle, which was consistent with the improvement effect of natural mutant cattle.

Blood biochemical parameters are of guiding significance in understanding the physiological state of animals. A study on fat-1 transgenic pigs demonstrated that the physiological and biochemical parameters, hematological parameters, and reproductive performance of transgenic pigs were all within the normal range [27]. Similarly, other studies had shown that *MSTN* DNA vaccine could increase skeletal muscle mass and endurance in mice, with no significant change in serum biochemistry [28]. In the case of TLR4 transgenic sheep, blood biochemical parameters, and the histology of major organs showed no abnormalities when compared to non-transgenic sheep [29]. In the current study, the continuous monitoring of blood biochemistry from 9 to 24 months in F1 and F2 generation cattle revealed that most detected parameters were similar between different generations, indicating no significant difference in protein metabolism, liver function, pancreas function, body immunity, and blood ions. However, changes in blood glucose levels and lactate were observed. Blood glucose content exhibited a strong negative correlation with body weight, and significant differences in blood glucose levels were observed among generations. Elevated fasting blood glucose was one of the criteria used to characterize diseases [30]. Previous studies had shown that *MSTN* knockout activates the AMPK signaling pathway to regulate glucose metabolism by increasing the AMP/ATP ratio [31]. *MSTN* played a role in glucose metabolism by promoting glucose consumption, glucose uptake, and glycolysis, while inhibiting glycogen synthesis in skeletal muscle cells [32]. Insulin resistance accelerates the decline in skeletal mass and strength [33]. Insulin resistance tended to escalate with advancing age and an increase in fat mass. The deficiency of the muscle growth inhibitor could improve glucose metabolism and reduce fat accumulation in type II diabetic mice. Multiple linear regression analysis suggested associations between several blood parameters and age. Compared to control cattle, older *MSTN* gene-edited cattle exhibited lower serum glucose and insulin levels, indicating that *MSTN* inhibition might help maintain insulin sensitivity with age, in line with findings from mice studies [34].

The high proportion of lymphocytes is often caused by viral infections. *MSTN* gene-edited cattle may exhibit enhanced resistance to viruses. Furthermore, the number of lymphocytes was also correlated with the weight and age of animals. According to a recent study [35], younger and lighter pigs had higher lymphocyte counts. Similar observations have been made in cattle, where lymphocyte counts and proportions decrease with increasing age. Moreover, a notable decrease in lymphocyte levels has been noted in the blood of *MSTN* gene-edited cattle in comparison to CON cattle. This finding provides additional evidence that *MSTN* gene alteration, contributing to increased body weight in cattle, influences blood parameters. However, when gad-transduced lymphocytes were injected into NOD mice at a high multiplicity of infection (m.o.i), insulin and blood glucose levels decreased [36]. This could be attributed to the low level of gad in lymphocytes, which remains unchanged under low m.o.i conditions.

Lipoprotein plays a crucial role in the whole-body lipid cycle as fat is insoluble in water. Lipoproteins can be categorized into five types based on density/size: high-density lipoprotein (HDL), low-density lipoprotein (LDL), medium-density lipoprotein (IDL), very low-density lipoprotein (VLDL), and ultra-low density lipoprotein (ULDL). LDL, also known as “bad cholesterol”, transports cholesterol from the liver to other organs and its accumulation has been associated with an increased risk of heart disease/stroke. On the other hand, HDL, known as “good cholesterol”, is not typically evaluated like LDL in terms of outcomes [37,38]. Previous GWAS results have shown that FADS1-related glucose increased are significantly associated with elevated total cholesterol, LDL cholesterol, and HDL cholesterol levels [39]. These results were similar to the results of this study, although HDL was higher than that in hybrid cattle. It is known that the obese have increased the serum levels of insulin [40].

Approximately 99% of the calcium in the body, in the form of calcium phosphate and a small amount of calcium carbonate, exists in teeth and bones to maintain their rigidity and structure. Around 1% of the calcium exists in the blood, extracellular fluid, muscle, and other tissues, and participates in vascular contraction and relaxation, nerve conduction, gland secretion, and other functions. This 1% calcium is very important for the physiological function of the body [41]. In this experiment, compared with Luxi cattle, the blood calcium of F1 and F2 hybrid cattle increased significantly within the normal range. Studies showed that insulin-like growth factor 1 (IGF-1), as an activator of the *MSTN* gene promoter, increased promoter activity by increasing cytoplasmic calcium in the myogenic context [42]. The wild-type MSTN protein was treated with increased concentration, which led to a decrease in promoter activity, indicating that the MSTN promoter was under the negative control of MSTN protein. However, when the mutant MSTN protein from Piedmontese cattle was used, the C-terminal active domain of the protein showed a transition from cysteine to tyrosine, and the promoter activity was not significantly affected [43]. We speculated a likely mechanism for increasing calcium in hybrid cattle. *MSTN* gene-edited hybrid cattle blood has a low concentration of mutant MSTN protein, which could not negatively regulate the activation of the MSTN promoter. It might increase IGF-1 by increasing Ca concentration as an alternative.

Most blood parameters showed poor correlation with body weight (as per the data provided). Among the blood parameters that positively correlated with body weight, most were expressed at higher levels in F1 and F2 hybrid cattle. These results indicate that altering the body weight of hybrid cattle have an impact on certain blood parameters.

In brief, cattle, as large livestock, have long intervals between generations and long cycles through conventional breeding techniques. Using CRISPR/Cas9 gene-editing technology to delete a piece of the *MSTN* gene sequence, the growth and meat quality traits of cattle were significantly improved without affecting the physiological and biochemical health of cattle. The new strain have better growth performance, reproductive performance, meat quality, and roughage performance inherited from Simmental cattle and *MSTN*-edited Luxi cattle.

## 5. Conclusions

In conclusion, our study demonstrates that the hybrid cattle with *MSTN* gene-edited Luxi bulls have the potential to enhance growth and performance. The hybrid cattle exhibited significant improvements in body size indexes, particularly in body weight, cross height, and hip height, indicating a notable development of the hindquarters. The F2 cattle showed further increases in body size indexes, and their coat color reverted back to the characteristic yellow of Luxi cattle. Furthermore, the *MSTN* gene-edited bulls displayed prominent acromions, deep and wide chests, and double-muscled forequarters and hindquarters with well-defined muscle masses. Multiple linear regression analysis revealed that the height of the hindquarters played a crucial role in the weight gain of *MSTN* gene-edited crossbred cattle. Additionally, significant differences were observed in blood glucose, calcium, and low-density lipoprotein levels, and a notable decrease in serum insulin levels was observed at 24 months of age. Overall, our findings suggest that utilizing *MSTN* gene-edited Luxi bulls in breeding programs can lead to improved growth and performance in crossbred cattle, benefiting both farmers and consumers alike.

## Figures and Tables

**Figure 1 animals-13-03216-f001:**
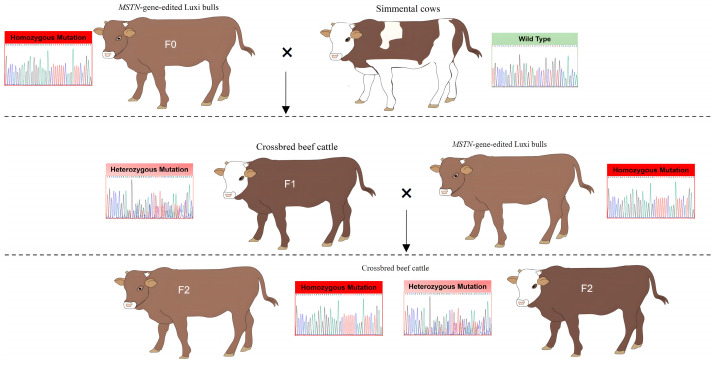
Flow chart of crossbred beef cattle. F0: *MSTN*^−/−^ Luxi bulls, F1 and F2: crossbred cattle of F1 and F2 generations. Homozygous Mutation (red): *MSTN*^−/−^, Heterozygous Mutation (pink): *MSTN*^+/−^, Wild Type (green): *MSTN*^+/+^.

**Figure 2 animals-13-03216-f002:**
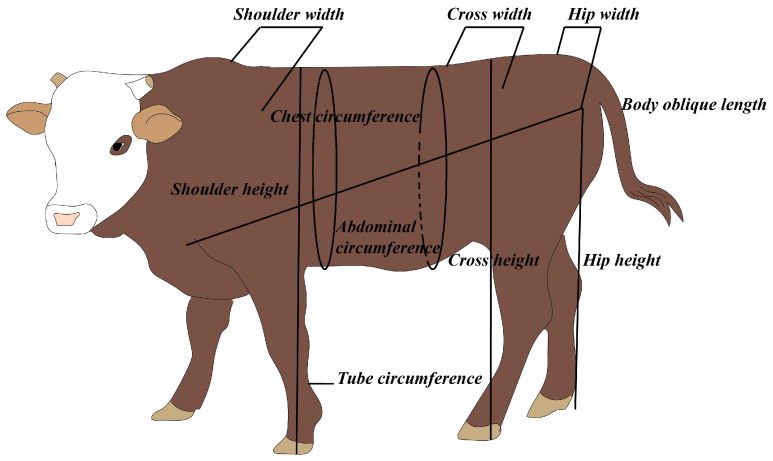
Schematic diagram of measurement of growth parameters.

**Figure 3 animals-13-03216-f003:**
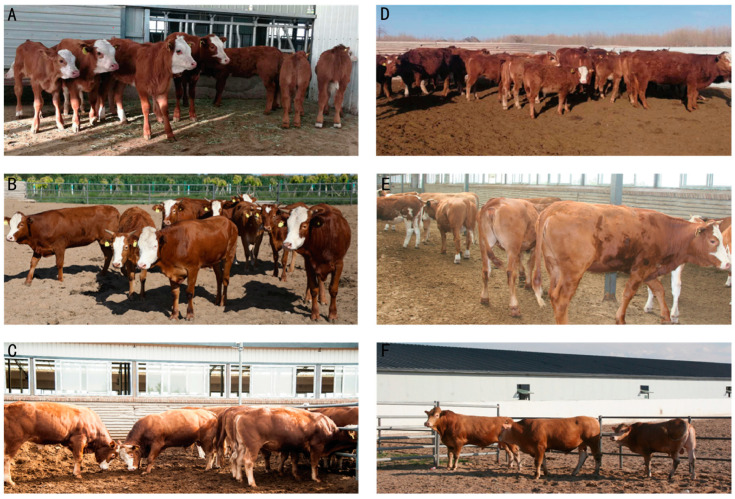
The hybrid cattle produced by *MSTN* gene-edited Luxi Yellow bulls and Simmental cows. (**A**–**C**): Calves, cows, and bulls of F1; (**D**–**F**): heifers, cows, and bulls of F2.

**Figure 4 animals-13-03216-f004:**
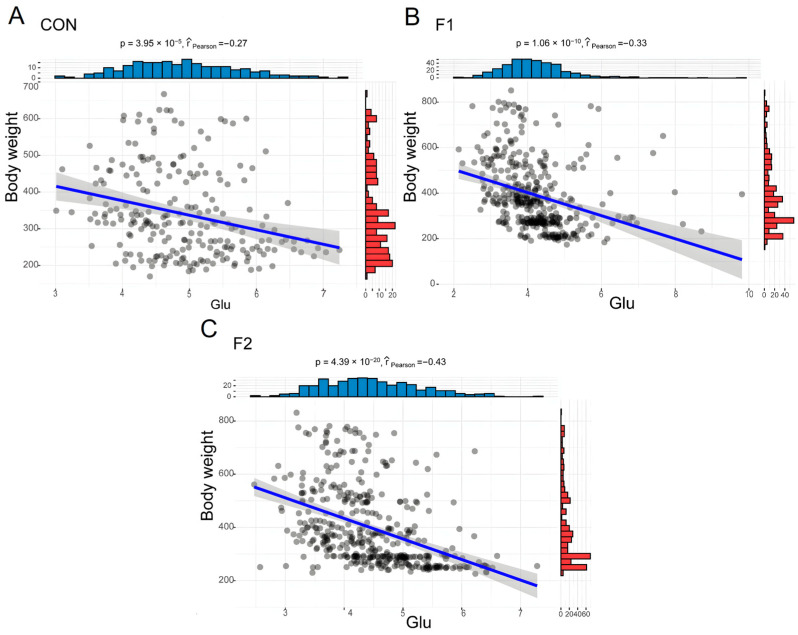
Linear relationship between blood glucose and BW in different generations of the hybrid cattle. (**A**) CON: Wild Type Luxi cattle. (**B**) F1: F1 generations of *MSTN* gene-edited hybrid cattle. (**C**) F2: F2 generations of *MSTN* gene-edited hybrid cattle. The black circles represent cattle at different ages, blue indicates the distribution of Glu in cattle of different ages, and red represents the distribution of body weight in cattle of different ages.

**Figure 5 animals-13-03216-f005:**
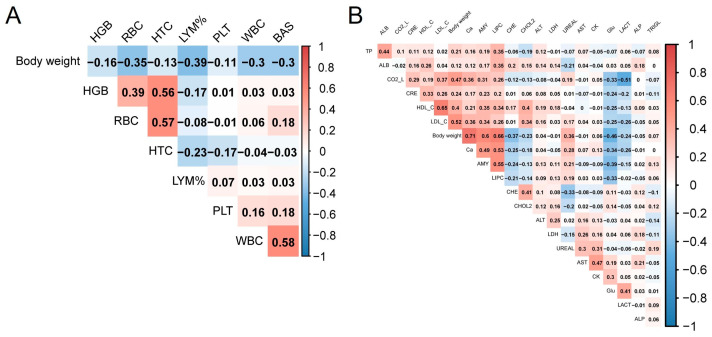
Relationship between BW and blood parameters. (**A**) Heatmap of correlations between BW and hematological parameters. (**B**) Heatmap of correlations between BW and biochemical parameters.

**Figure 6 animals-13-03216-f006:**
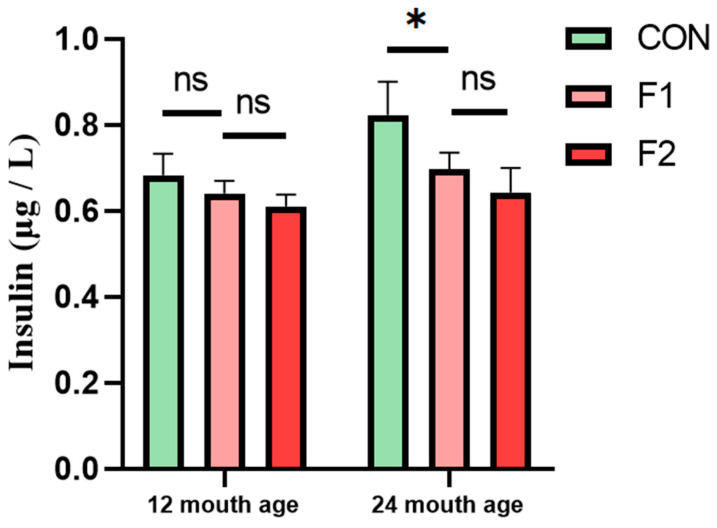
Serum insulin of different generations of cattle at 12 and 24 month, * *p* < 0.05. ns: no significance.

**Table 1 animals-13-03216-t001:** Effect of sex, age, and generation on growth parameters (average ± SD).

Parameters	Sex		Age								Generations			*p*-Value						
	M	F	0	3	6	9	12	15	18	24	WT	F1	F2	Sex	Age	Generation	Sex × Age	Sex × Generation	Age × Generation	Sex × Age × Generation
BW (kg)	305.7 ± 199.6 a	259.7 ± 150.2 b	41.1 ± 4 h	117.2 ± 4.8 g	182.5 ± 22 f	237.5 ± 26.2 e	276.5 ± 23.6 d	352.4 ± 30.9 c	439.6 ± 68.7 b	605 ± 106.1 a	252.9 ± 150.7 b	285.8 ± 184.1 a	293.3 ± 181.7 a	0.000	0.000	0.461	0.000	0.876	0.000	0.162
Shoulder height (cm)	116.5 ± 26 a	112.1 ± 21.8 b	72.1 ± 5.3 h	90.1 ± 2.2 g	102.6 ± 4.9 f	112.7 ± 5.5 e	124 ± 5.4 d	130.7 ± 6.7 c	134.4 ± 6.8 b	146.9 ± 7.1 a	112.9 ± 23.8 a	114.1 ± 23.8 a	115 ± 24.2 a	0.018	0.000	0.201	0.000	0.248	0.889	0.720
Cross height (cm)	119 ± 26.3 a	114.7 ± 22.8 b	74.1 ± 3.2 h	89.5 ± 2.2 g	105.5 ± 4.6 f	116.1 ± 5.8 e	128.1 ± 5.3 d	132.2 ± 6.9 c	138.6 ± 6.6 b	149.7 ± 6.1 a	114 ± 23.2 b	117.2 ± 24.7 ab	117.9 ± 25.2 a	0.013	0.000	0.028	0.000	0.024	0.751	0.484
Hip height (cm)	118.3 ± 26.1 a	114.7 ± 23.2 b	72.5 ± 4 h	89.9 ± 2.6 g	105.8 ± 3.4 f	115.7 ± 7.1 e	129.5 ± 5.9 d	132.4 ± 5.9 c	136.8 ± 6.2 b	148.7 ± 5.8 a	113.9 ± 23.7 b	116.8 ± 24.8 ab	117.5 ± 25 a	0.072	0.000	0.057	0.001	0.094	0.470	0.861
Shoulder width (cm)	35.7 ± 10.5 a	32.3 ± 8 b	20 ± 2.2 h	24.9 ± 2.2 g	28.2 ± 3.3 f	33.8 ± 3.3 e	38.2 ± 6.3 d	38 ± 4.5 c	40.3 ± 3.7 b	48 ± 5.3 a	32.9 ± 8.9 b	33.3 ± 9.2 b	35 ± 9.8 a	0.108	0.000	0.639	0.000	0.580	0.918	0.870
Cross width (cm)	37.9 ± 11.7 a	36 ± 8.8 b	21.4 ± 2.9 h	27 ± 2	30.4 ± 1.6 f	36.7 ± 3.9 e	39.3 ± 3.5 d	42.2 ± 4.1 c	45.6 ± 4.3 b	52.6 ± 6.2 a	35.7 ± 9.6 b	36.8 ± 10.4 ab	37.6 ± 10.6 a	0.004	0.000	0.971	0.000	0.927	0.880	0.527
Hip width (cm)	37.6 ± 11.9 a	35.4 ± 9.7 b	19.9 ± 3.2 h	25.6 ± 2.5 g	29.8 ± 1.7 f	36.1 ± 3.9 e	40.2 ± 3.7 d	41.4 ± 4 c	46 ± 4.7 b	52.5 ± 5.9 a	35.5 ± 10.4 a	36.3 ± 11 a	37.1 ± 11 a	0.000	0.000	0.749	0.000	0.857	0.177	0.720
Chest circumference (cm)	151.1 ± 44.9 a	146.3 ± 39.6 b	71.8 ± 3.3 h	109.7 ± 3 g	133.7 ± 3.9 f	148 ± 6 e	156.9 ± 6.3 d	173.4 ± 9.6 c	185.7 ± 16.1 b	209.1 ± 20.8 a	143.6 ± 39.6 b	149.9 ± 43.1 ab	150 ± 42.7 a	0.039	0.000	0.529	0.000	0.450	0.159	0.894
Abdominal circumference (cm)	167.8 ± 53.8 a	161.6 ± 45.6 b	72.7 ± 3.2 h	119.4 ± 4.2 g	145.8 ± 5.1 f	165.3 ± 9.7 e	178.8 ± 11.5 d	190.6 ± 20.5 c	210.1 ± 16.5 b	233.7 ± 14.3 a	162.6 ± 48.4 a	165.1 ± 49.3 a	165.1 ± 50.8 a	0.035	0.000	0.863	0.000	0.867	0.212	0.355
Body oblique length (cm)	124.6 ± 30.4 a	120 ± 26.1 b	74.2 ± 3.4 h	91.1 ± 4.2	110.7 ± 4 f	121.5 ± 7.4 e	131.4 ± 4.2 d	140.7 ± 8.9 c	144.8 ± 9.7 b	163.1 ± 10.1 a	120.5 ± 26.8 a	122.2 ± 28.1 a	123 ± 29.2 a	0.159	0.000	0.578	0.000	0.829	0.167	0.672
Tube circumference (cm)	17.4 ± 5.3 a	15.8 ± 3.6 b	10.1 ± 1.3 h	12.8 ± 1.1 g	14.2 ± 1.2 f	15 ± 1.2 e	16.6 ± 2.1 d	19.3 ± 2.9	21.1 ± 2.8 b	23 ± 2.4 a	16.6 ± 4.6 a	16.7 ± 4.7 a	16.3 ± 4.4 a	0.004	0.000	0.108	0.000	0.306	0.004	0.001

a–h indicate significance.

**Table 2 animals-13-03216-t002:** Effect of sex, age, and generation on hematological parameters (average ± SD).

Parameters	Sex		Age					Generation			*p*-Value						
	M	F	9	12	15	18	24	WT	F1	F2	Sex	Age	Generation	Sex × Age	Sex × Generation	Age × Generation	Sex × Age × Generation
WBC (×10 L^−9^)	8.51 ± 2.65	8.48 ± 2.50	9.87 ± 2.66 a	8.89 ± 2.52 b	8.14 ± 2.54 c	7.89 ± 2.41 c	7.72 ± 2.23 c	8.62 ± 2.57 a	8.36 ± 2.64 a	8.56 ± 2.58 a	0.007	0.000	0.624	0.091	0.045	0.266	0.325
LYM (%)	4.36 ± 2.80	4.75 ± 3.25	6.71 ± 4.32 a	4.75 ± 2.78 b	4.21 ± 2.2 bc	3.89 ± 1.44 c	2.91 ± 1.64 d	6.73 ± 3.24 a	3.66 ± 1.71 b	3.89 ± 3.01 b	0.517	0.000	0.000	0.139	0.834	0.000	0.269
RBC (×10 L^−12^)	8.01 ± 1.69	8.10 ± 1.78	9.08 ± 1.81 a	8.36 ± 1.3 b	7.86 ± 1.39 b	8.06 ± 1.77 b	6.85 ± 1.47 c	7.39 ± 1.36 b	8.28 ± 1.73 a	8.21 ± 1.81 a	0.026	0.000	0.013	0.174	0.001	0.021	0.023
HCT (%)	3229.3 ± 514.6	3384.5 ± 511.5	3286.59 ± 562.49 ab	3392.1 ± 393.59 a	3357.56 ± 475.39 a	3170.8 ± 552.82 b	3203.15 ± 553.41 b	3293.7 ± 384.96 b	3113.57 ± 521.03 c	3430.83 ± 541.38 a	0.551	0.210	0.068	0.339	0.167	0.007	0.073
PLT (×10 L^−9^)	297.26 ± 140.83	291.76 ± 156.65	385.84 ± 138.81 a	241.93 ± 96.79 c	261.6 ± 161.47 bc	286.56 ± 108.56 b	301.03 ± 168.56 b	283.91 ± 109.41 a	299.52 ± 127.76 a	298.65 ± 178.65 a	0.499	0.013	0.539	0.383	0.517	0.250	0.809
BAS (×10 L^−9^)	0.09 ± 0.08	0.08 ± 0.06	0.09 ± 0.07 a	0.11 ± 0.06 a	0.08 ± 0.04 b	0.08 ± 0.11 ab	0.07 ± 0.06 b	0.07 ± 0.04 b	0.1 ± 0.1 a	0.08 ± 0.06 b	0.962	0.291	0.945	0.628	0.797	0.965	0.431
HGB (g/dL)	100.18 ± 14.04	105.07 ± 13.68	106.66 ± 15.81 a	102.01 ± 14.11 b	103.18 ± 13.06 ab	97.65 ± 13.2 c	99.7 ± 12.42 bc	101.75 ± 10.55 b	98.31 ± 13.49 c	105.16 ± 15.7 a	0.477	0.472	0.406	0.912	0.533	0.826	0.795

a, b, or c indicate significance.

**Table 3 animals-13-03216-t003:** Effect of sex, age, and generation on biochemical parameters (average ± SD).

Parameters	Sex		Age					Generations			*p*-Value						
	M	F	9	12	15	18	24	WT	F1	F2	Sex	Age	Generation	Sex × Age	Sex × Generation	Age × Generation	Sex × Age × Generation
Glu (mmol/L)	4.23 ± 0.74	4.77 ± 1	5.19 ± 0.88 a	4.75 ± 0.75 b	4.39 ± 0.86 c	4.15 ± 0.68 d	3.99 ± 0.86 d	4.94 ± 0.8 a	4.22 ± 0.97 c	4.47 ± 0.82 b	0.000	0.000	0.003	0.267	0.470	0.385	0.132
AST (U/L)	62.71 ± 29.15	66.05 ± 28.12	65.69 ± 26.29 a	63.31 ± 28.73 a	64.55 ± 28.12 a	64.65 ± 33.19 a	63.45 ± 26.59 a	63.04 ± 20.3 a	63.64 ± 26.12 a	65.76 ± 34.72 a	0.661	0.551	0.075	0.651	0.110	0.112	0.153
ALT (U/L)	26.34 ± 7.26	24.28 ± 7.35	24.59 ± 7.6 b	25.36 ± 7.23 ab	24.66 ± 6.99 ab	25.17 ± 7.75 ab	26.9 ± 7.04 a	24.7 ± 5.3 b	26.63 ± 7.21 a	24.53 ± 8.4 b	0.663	0.783	0.432	0.998	0.368	0.374	0.509
TP (g/L)	67.26 ± 6.72	72.48 ± 7.87	67.75 ± 6.07 b	71.29 ± 6.04 ab	70.3 ± 7.39 ab	68.02 ± 8.66 ab	71.61 ± 9.18 a	64.75 ± 7.23 b	71.49 ± 6.66 a	71.34 ± 7.66 a	0.982	0.010	0.163	0.000	0.348	0.159	0.696
ALB (g/L)	32.34 ± 4.31	34.13 ± 4.3	33.37 ± 3.3 ab	33.39 ± 3.65 ab	33.25 ± 4.34 ab	32.04 ± 5.15 b	33.99 ± 5 a	31.26 ± 3.76 a	34.87 ± 3.84 a	32.87 ± 4.66 a	0.081	0.005	0.004	0.000	0.051	0.012	0.758
CK (U/L)	202.58 ± 143.25	218.07 ± 112.46	197.81 ± 125.04 a	212.84 ± 127.42 a	220.59 ± 124.5 a	205.89 ± 136.12 a	213.37 ± 132.66 a	202.13 ± 59.42 a	217.07 ± 189.2 a	208.57 ± 84.93 a	0.702	0.760	0.701	0.953	0.403	0.620	0.396
HDL-C (mmol/L)	3.27 ± 0.65	3.32 ± 1.22	2.91 ± 0.63 c	3.13 ± 0.57 b	3.49 ± 0.59 a	3.33 ± 1.77 ab	3.6 ± 0.51 a	3 ± 0.54 b	3.32 ± 0.72 ab	3.45 ± 1.29 a	0.002	0.094	0.073	0.015	0.071	0.096	0.188
LDL-C (mmol/L)	1.58 ± 0.44	1.6 ± 0.38	1.3 ± 0.4 d	1.41 ± 0.39 c	1.65 ± 0.33 b	1.72 ± 0.39 b	1.85 ± 0.29 a	1.67 ± 0.28 a	1.52 ± 0.51 c	1.6 ± 0.37 b	0.000	0.000	0.000	0.000	0.000	0.000	0.000
AMY (U/L)	28.21 ± 12.37	29.72 ± 12.86	19.68 ± 11.52 d	22.76 ± 10.14 c	31.31 ± 9.12 b	34.54 ± 12.63 a	36.43 ± 10.15 a	26.07 ± 10.95 b	30.61 ± 10.57 a	29.18 ± 14.86 a	0.446	0.000	0.814	0.111	0.643	0.664	0.409
Ca (mmol/l)	2.83 ± 0.35	2.96 ± 0.49	2.49 ± 0.22 d	2.58 ± 0.25 c	3.01 ± 0.4 b	3.2 ± 0.31 a	3.2 ± 0.31 a	2.83 ± 0.39 b	2.86 ± 0.4 b	2.96 ± 0.47 a	0.000	0.000	0.000	0.000	0.000	0.001	0.000
ALP (U/L)	143.06 ± 62.69	150.09 ± 61.97	156.31 ± 59.79 a	144.48 ± 63.18 a	149.41 ± 63.46 a	144.5 ± 61.61 a	137.67 ± 62.57 b	134.88 ± 57.63 b	151.49 ± 68.34 a	148.99 ± 58.56 a	0.714	0.650	0.392	0.302	0.639	0.401	0.427
LDH (U/L)	1076.97 ± 164.15	1093.3 ± 213.98	1131.2 ± 190.47 a	1060.43 ± 197.77 b	1066.09 ± 143.67 b	1102.58 ± 191.08 ab	1064.19 ± 210.75 b	1045.22 ± 118.53 b	1158.09 ± 187.09 a	1041.62 ± 207.55 b	0.014	0.040	0.255	0.032	0.013	0.113	0.028
CO2-L (mmol/L)	20.27 ± 3.88	21.79 ± 4.06	17.24 ± 3.48 b	20.59 ± 3.51 b	22.01 ± 3.33 a	22.24 ± 3.31 a	22.96 ± 3.78 a	21.36 ± 3.71 a	21.21 ± 3.96 ab	20.61 ± 4.27 b	0.000	0.833	0.012	0.003	0.003	0.030	0.012
CHE (U/L)	135.83 ± 30.47	123.13 ± 31.39	143.93 ± 28.9 a	134.59 ± 26.73 b	138.69 ± 31.96 ab	118.18 ± 29.51 c	112.93 ± 28.59 c	135.15 ± 30.79 a	126.65 ± 33.45 b	129.07 ± 29.74 b	0.397	0.010	0.018	0.921	0.190	0.143	0.469
CHOL (mmol/L)	2.09 ± 0.74	1.85 ± 0.68	2.27 ± 0.71 a	1.89 ± 0.66 b	1.85 ± 0.64 b	2.04 ± 0.87 b	1.81 ± 0.6 c	2.2 ± 0.58 a	1.96 ± 0.85 b	1.84 ± 0.62 b	0.816	0.473	0.065	0.825	0.166	0.707	0.783
LACT (mmol/L)	3.38 ± 2.18	3.83 ± 2.79	4.37 ± 2.2 a	3.91 ± 2.48 a	3.38 ± 2.77 ab	3.2 ± 2.12 b	3.12 ± 2.67 b	3 ± 1.75 b	3.98 ± 3.29 a	3.61 ± 1.93 a	0.552	0.506	0.116	0.949	0.369	0.285	0.388
TRIGL (mmol/L)	0.38 ± 0.14	0.41 ± 0.14	0.35 ± 0.13 b	0.42 ± 0.13 a	0.39 ± 0.14 ab	0.43 ± 0.15 a	0.39 ± 0.14 ab	0.43 ± 0.14 a	0.37 ± 0.1 c	0.4 ± 0.17 b	0.707	0.570	0.278	0.670	0.601	0.735	0.858
UREAL (mmol/L)	5.6 ± 1.82	4.98 ± 1.87	3.69 ± 1.39 d	5.16 ± 1.45 c	5.65 ± 1.48 b	5.6 ± 1.78 bc	6.38 ± 2.02 a	5.75 ± 1.38 a	4.86 ± 1.74 c	5.42 ± 2.14 c	0.021	0.966	0.050	0.000	0.018	0.140	0.009
LIPC (U/L)	19.94 ± 8.92	23.53 ± 9.81	14.52 ± 9.1 c	18.95 ± 9.61 b	23.73 ± 9.64 a	25.34 ± 6.58 a	25.86 ± 7.02 a	16.61 ± 6.9 c	24.26 ± 7.18 a	22.43 ± 11.46 b	0.550	0.000	0.263	0.014	0.977	0.694	0.745
CRE (μmol/L)	99.17 ± 25.49	103.85 ± 27.72	89.94 ± 22.11 c	100.69 ± 24.59 b	105.76 ± 24.09 ab	98.72 ± 26.89 b	112.08 ± 29.91 a	92.56 ± 13.68 c	109.35 ± 35.96 a	99.59 ± 19.73 b	0.670	0.261	0.715	0.233	0.321	0.803	0.205

a, b, c or d indicate significance.

## Data Availability

All the data generated or analyzed during the present study are available from the author upon reasonable request.

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
