# Peer review of "Effects of Different Generations and Sex on Physiological, Biochemical, and Growth Parameters of Crossbred Beef Cattle by Myostatin Gene-Edited Luxi Bulls and Simmental Cows"

_animals, 2023, doi:10.3390/ani13203216_

Round 1

Reviewer 1 Report

The authors investigated whether utilizing MSTN gene-edited Luxi bulls in breeding programs can lead to improved growth and performance in crossbred cattle. This research is very meaningful. However, some small details need to be revised and I recommend it to be accepted after a minor revision.

Line 143-165: The random residuals need to be added to all models, and the authors need to explain the reasons for considering these fixed effects and interactions in the model because the final model fit is not very good according to Figure 4;

Line 177: The cross height/width, and hip height of F1 cattle does not show significantly improvement compared with the control Luxi cattle, and the author needs to rephrase the statement;

Line 179: Is it necessary to indicate the specific traits, because some are not significantly different?

Line 183: There is no significant difference between age and generation, and the significant one is age & sex.

Line 184: Written error and the correct p-value is 0.028;

Line 185: “notable” was not suitable here because the no significant difference of hip height, also, the correct p-value here should be 0.057;

Line 188: Just cross height is significant difference and the author needs to rephrase the statement;

Line 199: HCT or HIT??

Line 206: This sentence needs to be stated in the discussion section “LYM (%) decreased significantly in generations and ages, which might be related to weight gain”.

Minor editing of English language required

Author Response

Response to Reviewer 1 Comments

1. Summary

2. Questions for General Evaluation

Reviewer’s Evaluation

Response and Revisions

Does the introduction provide sufficient background and include all relevant references?

Yes

We thank the reviewer for the positive comments.

Are all the cited references relevant to the research?

Yes

We thank the reviewer for the positive comments.

Is the research design appropriate?

Yes

We thank the reviewer for the positive comments.

Are the methods adequately described?

Yes

We thank the reviewer for the positive comments.

Are the results clearly presented?

Can be improved

Thank you for the reviewer's suggestion. We have made revisions to clarify the ambiguous results.

Are the conclusions supported by the results?

Yes

We thank the reviewer for the positive comments.

3. Point-by-point response to Comments and Suggestions for Authors

Comments 1: Line 143-165: The random residuals need to be added to all models, and the authors need to explain the reasons for considering these fixed effects and interactions in the model because the final model fit is not very good according to Figure 4;

Response 1: Thank you for your detailed review. We appreciate your feedback, and we apologize for any confusion caused. We have made the necessary modification to the expression of the multivariate linear regression model in the article.

In the article it was changed to the correct representation of the multivariate linear Regression model.

Regarding the reasons for considering these fixed effects and interactions, we thank you for bringing up this question. Firstly, in light of our experimental process and prior research findings, it has been observed that growth parameters, hematological parameters, and biochemical parameters exhibit differences among different gender and age groups, which are considered as important factors in blood-related variations and growth--related variations [1-3]. Secondly, the consideration of generations is based on the significant changes observed in the external appearance of hybrid cattle during breeding. Additionally, we wanted to understand how the MSTN gene changes with successive generations, and its impact on the hybrid cattle’s characteristics.

Regarding the consideration of interactions among these fixed effects, we initially conducted analyses using non-interacting models and compared them with models that incorporated fixed effects and interactions. We found that several predictors possibly exhibited synergistic effects with each other, such as age and gender or age and generation, as shown in Table 1, 2, and 3. The inclusion of sex, age, and gene interaction terms was found to have a significant impact on certain growth, hematological, and biochemical variables. Moreover, based on the ANOVA analysis in R, the models that incorporated fixed effects and interactions demonstrated better overall model fit, as more parameters showed improved goodness of fit. Consequently, we decided to include these fixed effects and interactions in the final model to capture the complex relationship between sex, age, generation, and the observed changes in MSTN gene-edited cattle.

In response to your concern about the model fit in Figure 4, we acknowledge that it might appear less satisfactory due to the significant influence of age on growth parameters in cattle. The age are known to display stage variations at different stages, which could contribute to a relatively lower model fit. Nonetheless, we believe that the model provides valuable insights into the characterization of MSTN gene editing effects overall. Thank you for emphasizing this crucial aspect once again.

Comments 2: Line 177: The cross height/width, and hip height of F1 cattle does not show significantly improvement compared with the control Luxi cattle, and the author needs to rephrase the statement

Response 2: Thank you for your detailed review. We have made the necessary modifications accordingly. In the F2 generation of cattle, there was a significant improvement in body weight, cross height/width, and hip height, indicating robust development of the hind limb muscles. This section has also been revised in the manuscript.

Comments 3: Line 179: Is it necessary to indicate the specific traits, because some are not significantly different?

Response 3: This is a great question. We actually had similar considerations during the earlier stages of our study, and we conducted Stepwise Regression analysis to select the most significant variables. By doing so, we aimed to describe the MSTN gene-edited cattle using the minimum number of traits that exhibited the most pronounced changes.

However, in this study, we chose to include all the variables to comprehensively explore the overall changes in MSTN gene-edited cattle across different age groups, genders, and generations. This approach allowed us to gain a comprehensive understanding of the effects of MSTN gene editing and determine which variables had significant impacts and which ones did not.

By including all the variables, we aimed to provide comprehensive guidance and insight into the role of MSTN gene editing, not only for cattle but also for other animals. This approach enabled us to identify the specific parameters that were significantly affected by MSTN gene editing while also recognizing those that did not exhibit significant changes.

Comments 4: Line 183: There is no significant difference between age and generation, and the significant one is age & sex.

Response 4: Thank you for your thorough review. We appreciate you pointing out the error in Line 183 of the manuscript. We have made the necessary correction and found that there is indeed no significant difference between age and generation. However, we did observe a significant association between age and sex. We apologize for any confusion caused by the initial statement and have revised it accordingly.

Comments 5: Line 184: Written error and the correct p-value is 0.028;

Response 5: Thank you for your thorough review. We have made the necessary correction for the written error in Line 184 of the manuscript. The correct p-value is indeed 0.028. We appreciate you bringing this to our attention and ensuring the accuracy of the reported results.

Comments 6: Line 185: “notable” was not suitable here because the no significant difference of hip height, also, the correct p-value here should be 0.057;

Response 6: Thank you for your suggestion. We have made the corresponding modification regarding the ambiguity caused by the word "notable". The description of p=0.057 has been revised to "there is a moderate difference observed in hip height, although non-significant".

Comments 7: Line 188: Just cross height is significant difference and the author needs to rephrase the statement;

Response 7: Thank you for your feedback. Based on the information you have provided, there appears to be a problem with the statement on line 188 and it needs to be reworded. We have made the necessary changes and have marked it in the appropriate place.

Comments 8: Line 199: HCT or HIT??

Response 8: We sincerely apologize for the confusion caused by our oversight. We have realized that there was an ambiguity in the abbreviation “HCT” in the manuscript, which should stand for hematocrit. We have rectified this error in Table 2 to ensure accuracy.

We greatly appreciate the reviewer’s correction and sincerely apologize for any confusion it may have caused. We value your feedback and suggestions to ensure the accuracy and clarity of the manuscript. Thank you very much for your understanding and support.

Comments 9: This sentence needs to be stated in the discussion section “LYM (%) decreased significantly in generations and ages, which might be related to weight gain”.

Response 9: Thank you for your suggestion. Based on the suggestion, we have included the discussion regarding the significant decrease in LYM (%) in different generations and age groups in our manuscript. We have also linked this finding to previous research on the relationship between LYM and animal weight and age. By doing so, we have provided supporting evidence for the correlation between the increased weight in MSTN gene-edited cattle and the changes observed in LYM.

In particular, our study revealed a significant downward trend in LYM (%) across generations and age groups. This finding suggests that there might be a relationship between LYM and weight gain in the MSTN gene-edited cattle. Previous studies have reported that LYM levels tend to decrease as animals grow older and gain weight [4]. In line with these findings, the observed decrease in LYM (%) in our study further supports the notion that the increased weight in MSTN gene-edited cattle is associated with LYM changes.

It is worth noting that the underlying mechanisms linking LYM and weight gain in MSTN gene-edited cattle require further investigation. Future research can delve into the molecular pathways and biological processes associated with LYM regulation and its impact on weight gain. Nevertheless, our study provides initial evidence for the correlation between weight gain and LYM changes in MSTN gene-edited cattle. Related discussion added to Line 336-343 of the manuscript.

4. Response to Comments on the Quality of English Language

Point 1: Minor editing of English language required

Response 1: Thank you for your feedback on the quality of English language in our manuscript. We apologize for any errors or shortcomings that may have been present. We appreciate your recommendation for minor editing to improve the English language.

In response to your suggestion, we have carefully reviewed the manuscript and made necessary revisions to enhance the clarity and accuracy of the language. We have also utilized language editing tools to improve grammar, syntax, and sentence structure. Additionally, we have enlisted the assistance of professional proofreaders to ensure the highest standard of English language in the final version of the manuscript.

We are grateful for your attention to detail and your valuable input in this regard. If you have any further specific suggestions or concerns, please don't hesitate to let us know. We are committed to delivering a manuscript that meets the highest linguistic standards.

5. Additional clarifications

NO

Reference

  1. Tornador C, Sánchez-Prados E, Cadenas B, Russo R, Venturi V, Andolfo I, et al. Codysan: A telemedicine tool to improve awareness and diagnosis for patients with congenital dyserythropoietic anemia. Front Physiol. 2019; 10:1063. DOI: 10.3389/fphys.2019.01063.
  2. Barker EN, Tasker S, Day MJ, Warman SM, Woolley K, Birtles R, et al. Development and use of real-time pcr to detect and quantify mycoplasma haemocanis and "candidatus mycoplasma haematoparvum" in dogs. Vet Microbiol. 2010; 140(1-2):167-70. DOI: 10.1016/j.vetmic.2009.07.006.
  3. Nollens HH, Rivera R, Palacios G, Wellehan JF, Saliki JT, Caseltine SL, et al. New recognition of enterovirus infections in bottlenose dolphins (tursiops truncatus). Vet Microbiol. 2009; 139(1-2):170-5. DOI: 10.1016/j.vetmic.2009.05.010.
  4. Pabst R. The pig as a model for immunology research. Cell Tissue Res. 2020; 380(2):287-304. DOI: 10.1007/s00441-020-03206-9.

Reviewer 2 Report

The authors of the article presented an interesting study for the practical use of crossbred beef cattle by myostatin gene-edited Luxi bulls and Simmental cows.

There are some comments that require correction:

1. Whole genome sequencing data from the edited genome must be submitted to provide evidence of the quality of the edited animal.

2. There is no information about the living conditions of the animals. Was the comparison carried out at the same time?

3. P-value are not reported in the correlation analysis.

4. You write on line 285 that "Two of these genes, CDKAL1 and E2F3, demonstrated associations with specific single nucleotide polymorphisms (SNPs) that played a role in muscle development regulation [26]". Why were genes CDKAL1 and E2F3 chosen for discussion? It is necessary to either justify why these genes are given, or remove them from the discussion.

Author Response

Response to Reviewer 2 Comments

1. Summary

2. Questions for General Evaluation

Reviewer’s Evaluation

Response and Revisions

Does the introduction provide sufficient background and include all relevant references?

Yes

We thank the reviewer for the positive comments.

Are all the cited references relevant to the research?

Can be improved

Thanks to the reviewers' suggestions, we have revised and improved the references.

Is the research design appropriate?

Yes

We thank the reviewer for the positive comments.

Are the methods adequately described?

Can be improved

Thanks to the reviewers' suggestions, we have added to the description of the methodology.

Are the results clearly presented?

Yes

We thank the reviewer for the positive comments.

Are the conclusions supported by the results?

Yes

We thank the reviewer for the positive comments.

3. Point-by-point response to Comments and Suggestions for Authors

Comments 1: Whole genome sequencing data from the edited genome must be submitted to provide evidence of the quality of the edited animal.

Response 1: First of all, thank you very much for your valuable feedback on the manuscript. We would like to address the issue raised by the reviewer regarding the need for whole genome sequencing data from the edited animal to provide evidence of the quality of the edited animal.

The MSTN gene-edited cattle in our study were generated using the Crispr/Cas9 technology [1]. The founder bull underwent comprehensive identification after birth to ascertain the information of the MSTN gene locus, which served as the basis for establishing the new strain of MSTN gene-edited cattle. Additionally, we conducted various analyses to demonstrate the reliability of gene editing in the MSTN gene-edited cattle [2-8]. For subsequent hybrid offspring, blood samples were collected at birth for DNA extraction to perform PCR genotyping analysis. The following are partial results of MSTN gene identification in the F1 generation of cattle, which fully validate the quality and reliability of our MSTN gene-edited cattle. Relevant additional results have been added to the supplementary material.

Supplementary Table 1 The genotyping information of F1 generation.

Genotype

Sex

Number of cattle

All

MSTN-g505del(115)

Male

17

44

Female

27

MSTN-g507del(6)

Male

13

31

Female

18

Supplementary Figure 1. Generation and identification of MSTN gene-edited cattle. (A) The schematic diagram of MSTN gene and the editing sites; (B) Sequencing chromas for MSTN-g.507del (6) in the MSTN gene; (C) Sequencing chromas for MSTN-g.505del (115) in the MSTN gene.

These data and results showcase the accuracy and reliability of the genotypes of the MSTN gene-edited cattle in our study. While we have not conducted whole genome sequencing yet, these genotyping results serve as substantial evidence of the quality of our MSTN gene-edited cattle. We will consider incorporating whole genome sequencing in future research to provide further evidence of the edited genome’s quality.

Comments 2: There is no information about the living conditions of the animals. Was the comparison carried out at the same time?

Response 2: Thank you for your question. Considering that our MSTN gene-edited Luxi cattle were generated using the Crispr/Cas9 system, the timing of different generations is not the same. However, for the same generation of cattle, we selected hybrid cattle within ±1 month of age for tracking measurements and blood tests. This part has been added in 2.1. Animals. Additionally, our hybrid cattle were raised in a professional beef cattle farming enterprise, Sheng-Quan Ecological Animal Husbandry Company, which has a well-established and mature feeding system. This ensures that the batches of cattle and different generations have almost consistent feeding and management levels. Feed composition and nutrient composition of refined feed for normal adult cattle at different stages are provided in the table below. Relevant additional results have been added to the supplementary material.

Supplementary Table 2 Feed composition

Feed composition

Quality(kg)/day

Silage (kg/head)

12

Gluten (kg/head)

2

Hay (bale/head)

2

Refined feed

2.5

Supplementary Table 3 Nutrient composition of refined feed

Nutrients

Composition

Crude protein, not less than

16.0

Crude fat, not more than

12.0

Crude fiber, not more than

9.0

Calcium

0.5-1.8

Total phosphorus, not less than

0.4

Sodium chloride

0.8-1.5

Lysine, not less than

0.4

Comments 3: P-value are not reported in the correlation analysis.

Response 3: Thank you to the reviewer for your valuable suggestion. Due to formatting constraints, the initial version of the manuscript did not provide the P-values for the correlation analysis. However, we have now included the relevant data, including the correlation coefficients and corresponding P-values, in the Supplementary Table 4. Additionally, in section 3.4 of the manuscript titled "Relationship between BW and Hematological, and Biochemical Parameters," we have provided a supplementary explanation regarding these statistical values. Relevant additional results have been added to the supplementary material.

Comments 4: You write on line 285 that "Two of these genes, CDKAL1 and E2F3, demonstrated associations with specific single nucleotide polymorphisms (SNPs) that played a role in muscle development regulation [26]". Why were genes CDKAL1 and E2F3 chosen for discussion? It is necessary to either justify why these genes are given, or remove them from the discussion.

Response 4: Thank the reviewer for your suggestion. We appreciate the thorough evaluation of our manuscript. The reasons for including the CDKAL1 and E2F3 genes in the discussion section are the following. Firstly, they may potentially be associated with the MSTN gene. Through GWAS and replication studies, CDKAL1 and KLF9 have been identified as novel loci associated with body mass index (BMI) in East Asians, thereby influencing the BMI of this population [9]. Furthermore, a significant gene-gene interaction for BMI between the KLF9 and GDF8 loci has been observed. Secondly, CDKAL1 has been studied and shown to be an SNP locus related to muscle development in Simmental cattle [10]. Additionally, E2F3 is a major regulator that specifically targets genes involved in muscle progenitor cells [11], indicating a potential association with the MSTN gene. However, considering that a comprehensive discussion of these genes would require the inclusion of multiple references, it might divert the attention from the main conclusions of our manuscript. Therefore, we have decided to remove this part from the discussion section.

In conclusion, we sincerely appreciate your attention to this matter and your valuable suggestions. If you have any further questions or comments, please feel free to let us know.

4. Response to Comments on the Quality of English Language

Point 1: I am not qualified to assess the quality of English in this paper.

Response 1: Thank you for your feedback. We understand that your expertise lies in assessing the scientific content rather than the quality of English in the paper. We apologize for any confusion caused by our previous response. We appreciate your valuable insights regarding the scientific aspects of the paper, and we will address those points accordingly. If you have any further questions or concerns regarding the scientific content, please let us know.

5. Additional clarifications

NO

Reference

  1. Zhao Y, Yang L, Su G, Wei Z, Liu X, Song L, et al. Growth traits and sperm proteomics analyses of myostatin gene-edited chinese yellow cattle. Life (Basel). 2022; 12(5) DOI: 10.3390/life12050627.
  2. Gao L, Yang M, Wang X, Yang L, Bai C, Li G. Mstn knockdown decreases the trans-differentiation from myocytes to adipocytes by reducing jmjd3 expression via the smad2/smad3 complex. Biosci Biotechnol Biochem. 2019; 83(11):2090-96. DOI: 10.1080/09168451.2019.1644152.
  3. Sheng H, Guo Y, Zhang L, Zhang J, Miao M, Tan H, et al. Proteomic studies on the mechanism of myostatin regulating cattle skeletal muscle development. Front Genet. 2021; 12:752129. DOI: 10.3389/fgene.2021.752129.
  4. Gu M, Wei Z, Wang X, Gao Y, Wang D, Liu X, et al. Myostatin knockout affects mitochondrial function by inhibiting the ampk/sirt1/pgc1α pathway in skeletal muscle. Int J Mol Sci. 2022; 23(22) DOI: 10.3390/ijms232213703.
  5. Wang X, Wei Z, Gu M, Zhu L, Hai C, Di A, et al. Loss of myostatin alters mitochondrial oxidative phosphorylation, tca cycle activity, and atp production in skeletal muscle. Int J Mol Sci. 2022; 23(24) DOI: 10.3390/ijms232415707.
  6. Wu D, Gu M, Wei Z, Bai C, Su G, Liu X, et al. Myostatin knockout regulates bile acid metabolism by promoting bile acid synthesis in cattle. Animals (Basel). 2022; 12(2) DOI: 10.3390/ani12020205.
  7. Zhu L, Wang X, Wei Z, Yang M, Zhou X, Lei J, et al. Myostatin deficiency enhances antioxidant capacity of bovine muscle via the smad-ampk-g6pd pathway. Oxid Med Cell Longev. 2022; 2022:3497644. DOI: 10.1155/2022/3497644.
  8. Wu D, Wang S, Hai C, Wang L, Pei D, Bai C, et al. The effect of mstn mutation on bile acid metabolism and lipid metabolism in cattle. Metabolites. 2023; 13(7) DOI: 10.3390/metabo13070836.
  9. Okada Y, Kubo M, Ohmiya H, Takahashi A, Kumasaka N, Hosono N, et al. Common variants at cdkal1 and klf9 are associated with body mass index in east asian populations. Nat Genet. 2012; 44(3):302-6. DOI: 10.1038/ng.1086.
  10. Bordbar F, Mohammadabadi M, Jensen J, Xu L, Li J, Zhang L. Identification of candidate genes regulating carcass depth and hind leg circumference in simmental beef cattle using illumina bovine beadchip and next-generation sequencing analyses. Animals (Basel). 2022; 12(9) DOI: 10.3390/ani12091103.
  11. Julian LM, Liu Y, Pakenham CA, Dugal-Tessier D, Ruzhynsky V, Bae S, et al. Tissue-specific targeting of cell fate regulatory genes by e2f factors. Cell Death Differ. 2016; 23(4):565-75. DOI: 10.1038/cdd.2015.36.
